



# Multi-hazard susceptibility mapping of cryospheric hazards in a high-Arctic environment: Svalbard Archipelago

Ionut C. Nicu[1,2], Letizia Elia[3], Lena Rubensdotter[4,5], Hakan Tanyaş[6], Luigi Lombardo[6]

[1]High North Department, Norwegian Institute for Cultural Heritage Research (NIKU), Fram Centre, N-9296, Tromsø, Norway
[2]College of Humanities, Arts and Social Sciences, Flinders University, Adelaide, SA 5042, Australia
[3]Department of Physics and Astronomy, University of Bologna, Viale Berti Pichat 6/2, 40127 Bologna, Italy
[4]Geological Survey of Norway (NGU), P.O. Box 6315 Torgarden, 7491 Trondheim, Norway
[5]Arctic Geology Department, The University Centre in Svalbard (UNIS), P.O. Box 156, 9171, Longyearbyen, Norway
[6]Faculty of Geo-Information Science and Earth Observation (ITC), University of Twente, PO Box 217, Enschede, AE 7500, Netherlands

*Correspondence to*: Ionut C. Nicu (ionut.cristi.nicu@niku.no) and Letizia Elia (letizia.elia2@unibo.it)

**Abstract.** The Svalbard Archipelago represents the northernmost place on Earth where cryospheric hazards, such as thaw slumps (TS) and thermo-erosion gullies (TEG) could take place and rapidly develop under the influence of climatic variations. Svalbard permafrost is specifically sensitive to rapidly occurring warming and therefore, a deeper understanding of TS and TEG is necessary to understand and foresee the dynamics behind local cryospheric hazards' occurrences and their global implications. We present the latest update of two polygonal inventories where the extent of TS and TEG is recorded across Nordenskiöld Land (Svalbard Archipelago), over a surface of approximately 4000 km[2]. This area was chosen because it represents the most concentrated ice-free area of the Svalbard Archipelago and, at the same time, where most of the current human settlements are concentrated. The inventories were created through visual interpretation of high-resolution aerial photographs, as part of our ongoing effort toward creating a pan-Arctic repository of TS and TEG. Overall, we mapped 562 TS and 908 TEG, from which we separately generated two susceptibility maps using a Generalized Additive Modelling (GAM) approach, under the assumption that TS and TEG manifest across Nordenskiöld Land, according to a Bernoulli probability distribution. Once validating the modelling results, the two susceptibility patterns were combined into the first multi-hazard cryospheric susceptibility map of the area. The two inventories are available at https://doi.org/10.1594/PANGAEA.945348 (Nicu et al., 2022a) and https://doi.pangaea.de/10.1594/PANGAEA.945395 (Nicu et al., 2022b).

## Short Summary

Thaw slumps and thermo-erosion gullies are cryospheric hazards that are widely encountered in Nordenskiöld Land, the largest and most compact ice-free area of Svalbard Archipelago. By statistically analysing the landscape characteristics of locations where these processes occurred, we can estimate where they may occur in the future. We mapped 562 thaw slumps



and 908 thermo-erosion gullies and used them to create the first multi-hazard susceptibility map in a high-Arctic
environment.

## 1 Introduction

Permafrost constitutes subsurface materials that remain continuously at or below 0°C for at least two consecutive years. The rapidly increasing temperatures recorded since the 1980s have initiated permafrost degradation in many Arctic regions (Smith et al., 2022; Biskaborn et al., 2019). The cryosphere (including sea ice, glaciers, lake and river ice, continental ice
sheets, seasonal snow, permafrost, and seasonally frozen ground) covers roughly 14% of the Earth's surface. Some atmospheric hazards such as hail, frost and freezing rain have globally decreased in recent years (Ding et al., 2021). In Arctic conditions, this effect implies a reduced ice cover forming over the underlying permafrost soil, which therefore in turn gets increasingly exposed to subaerial conditions (Gilbert et al., 2018). This mechanism is, together with prolonged seasons with +0 degrees, one of the main drivers of permafrost degradation. Permanent thawing of the internal ice in permafrost soils
often leads to subsidence and slumps, which are called thermokarst (Kokelj and Jorgenson, 2013).

Thermokarst is a significant threat in Arctic environments, and numerous examples of its negative effects have been reported at various scales, across several ecosystems (Voigt et al., 2019), infrastructure types (Hjort et al., 2018; Hjort et al., 2022), and affecting cultural heritage sites (Nicu et al., 2021a; Nicu et al., 2021b; Nicu et al., 2022c). Aside from these directly observable effects on the ground, permafrost thawing can also release greenhouse gases such as carbon dioxide and methane
into the atmosphere, thus contributing to global warming (Oberle et al., 2019; Ran et al., 2022). At the meso-scale, one of the consequences of warming permafrost ground consists in the deepening of the active layer. This layer represents the uppermost part of the soil column, subjected to seasonal thawing and refreezing. Therefore, as warming occurs, the part of the soil column where this cycle takes place becomes increasingly deep, whereas previously ice would have held the soil particles together at these depths (Frey and Mcclelland, 2009; Schaefer et al., 2011). In turn, this naturally results in reduced
cohesion between soil particles, something that can promote the initiation of geomorphic processes unique of Arctic environments, knows as thaw slumps (TS, also depending on water released from ground ice) (Cassidy et al., 2017) and thermo-erosion gullies (TEG) (Godin et al., 2012). The precise feed-back mechanisms involved in TS and TEG activity are still relatively poorly understood.

TS are caused by the thawing of ice-rich permafrost which, independently or together with precipitation, result in
oversaturated soils. This induces significant loss in terms of shear strength, and may lead to soil collapses, forming slumps (Daanen et al., 2012). TS can initiate along an erosive riverbank or shoreline, or even within a TEG, where fluvial erosion exposes ice-rich frozen ground to rapid thawing (Nicu et al., 2021a; Cassidy et al., 2017). Conversely, a TEG may be initiated in response to heat transfer along preferential directions. This is the case when water infiltrates into the soil column warming the surrounding material and causing loss of cohesion. This may occur in or along seasonal freeze-cracks in the
ground, sometimes in connection to ice-wedge polygons. Something that can also add further instability is the increase in the



active layer's depth due to the same heat transfer process. Below the active layer, the ground remains permanently frozen, with the upper portion being commonly referred to as transition zone (Godin and Fortier, 2012). This ice-rich transition zone will, if thawed, release excess water that may further initiate small scale fluvial processes and small slumps or grain collapses. TEG can develop both retrogressively upslope and through widening/deepening of the initial incision (Iwahana et al., 2014; Nicu et al., 2022c).

Over the last years, there has been an increasing interest in studies referring to TS activity in permafrost regions of China (Niu et al., 2015; Xia et al., 2021), Russia (Séjourné et al., 2015), Alaska (Swanson and Nolan, 2018; Swanson, 2021), Canada (Lewkowicz and Way, 2019), and Svalbard (Nicu et al., 2021a). TEG are less studied, except for a few cases in Canada (Godin et al., 2014; Godin et al., 2019), Russia (Sidorchuk, 2019), and Svalbard (Nicu et al., 2022c). Hardly any of

these research efforts though have focused on learning from past TS and TEG occurrences to estimate locations where they may form in the future (Yin et al., 2021). This concept, at lower latitudes and for other geomorphological processes is usually referred to as susceptibility, or the probability of a given process to occur across a given landscape (Hansen, 1984). However, single susceptibility maps would not be highly informative in an Arctic context where TS and TEG can take place within the same terrain and be mutually triggering. For this reason, a much more interesting scientific product would consist

of a multi-hazard susceptibility map where the likelihood of TS and TEG is combined to highlight locations where these processes may contextually initiate and develop.

Multi-hazard assessment is also part of the Agenda 21 for Sustainable Development (Un Department of Economic and Social Affairs, 1992). Its relevance is highlighted in the context of risk reduction strategies because the combination of one or more hazards together (especially cryospheric ones) may be more threatening than the occurrence of one (Kappes et al.,

2012). Even aside the specific peri-Arctic context, multi-hazard susceptibility modelling is rarely touched upon, with few examples on landslides and gully erosion (Lombardo et al., 2020), rock fall and debris fall (Saha et al., 2021), floods, landslides, and gully erosion (Javidan et al., 2021). Specifically in the context of cryospheric hazards though, the current literature offers no examples in the Arctic.

Our work fits in this gap and aims to bring two essential elements to the attention of the geoscientific community. The first is

related to the limited availability of cryospheric hazard inventories, for which we try here to promote a positive habit of data sharing, a fundamental aspect of scientific progress especially when working in an unchartered territory such as the Arctic regions, local processes, and their manifestation in response to climate change. For this reason, we share the first update of two TS and TEG inventories mapped across the Nordenskiöld Land (Svalbard Archipelago), an area covering roughly 4000 km$^2$. The second objective of this work is to produce locally valuable probabilistic estimates of TS and TEG occurrences and

their multi-hazard relation. This is achieved by implementing two separate binomial Generalized Additive Models (GAM), whose results are explored in depth both by interpreting landscape characteristics associated with one or the other hazard under consideration and by validating the predictive patterns via a set of performance assessment tools.





## 2 Study area

Svalbard Archipelago covers an area of about 61,020 km² and is located halfway through the North Pole and the coast of Norway (Fig. 1a) (Zwoliński et al., 2013). The study area is located in central Spitsbergen (Fig. 1b), which represents the largest island of the Svalbard Archipelago (governed by Norway and established by the Spitsbergen Treaty from 9 February 1920). The average annual air temperature for Svalbard calculated for the 30 years between 1988 and 2017 was 1.5°C higher than the same for the reference period 1971-2000 (Hanssen-Bauer et al., 2019). In Svalbard, the projected temperatures' increase in the 21st century varies from a few percent in the SW to more than 40% in the NE (Førland et al., 2011). This increase in temperature is likely to be driven by sea ice decline, higher sea surface temperature, and a general background warming (Isaksen et al., 2016). As a result, the permafrost is expected to degrade even further in the future. Moreover, a significant increase in rainfall discharges has been locally recorded over the last century, with annual precipitation in 1940 measured at 482 mm and reaching 704 mm in 2018. The period between October and March corresponds to the wettest season (overlapping the period of high cyclonic activity), followed from April to July by the driest. Specifically, precipitation during winter is up to two times higher than in summertime (Demidov et al., 2021).

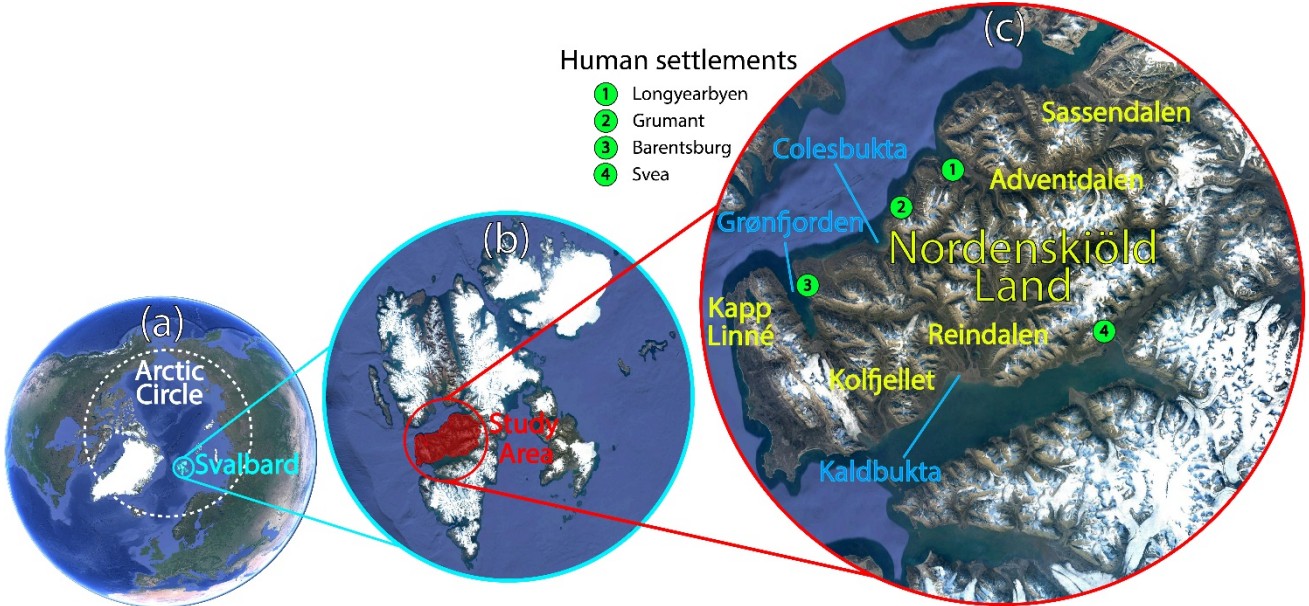

**Figure 1. Panels showing location of the study area in the context of a) the Northern Hemisphere; b) the Svalbard Archipelago and c) local settlements, with colour coded details where toponyms appear in yellow and fjords in blue (base maps from © Google Earth).**

Svalbard represents one of the most diverse geological landscapes in the world, where sections representing most of the Earth's history are accessible. Outcropping bedrock formations in Svalbard range from the Archaean to Quaternary in age and were uplifted during the Cenozoic era (Koevoets et al., 2019). Geologically, the peninsula is part of the contact zone of



two large structures of the first order: the horst-anticlinorium of the western coast of Spitsbergen and the West Spitsbergen graben. Quaternary deposits (soft sediment) consist of isostatically raised marine sediments in the lowlands, glacial and glacio-fluvial deposits in the valleys, extensive and complex slope deposits, areas of aeolian sediment cover and extensive in situ weathering of bedrock. The landscape is particularly diverse: from watershed peaks to the landscapes of U-shaped valleys, extensive mountain plateaus, small valley glaciers and moraines, and coastal plains. Much of the terrain hosts marked mountains surfaces, steep slopes and moraines draped by primary and desert-Arctic soils with thin herbaceous-moss-lichen groups. All sediments and bedrock are heavily influenced by the perennial frost in the ground (permafrost) (Demidov et al., 2021). Over time, especially the more fine-grained deposits have accumulated an excess of ground ice, especially the upper 1-5 m of the permanently frozen soil (Gilbert et al., 2018).

Nordenskiöld Land area was specifically chosen for this study, because it represents the largest and compact ice-free peninsula of Svalbard archipelago, located between Isfjorden, Van Mijenfjorden and Bellsund (Fig. 1c). It also represents the area where most of the human settlements (Longyearbyen – and recreational huts in the vicinity, Barentsburg, and Svea – a mining city whose activities may be decommissioned soon) and infrastructure are located. In addition, there is a lot of transport by snow mobile and dog sledging during the winter season, and on foot in this area for recreational and practical purposes. This makes the present study highly relevant from a societal point of view, considering that this century the Arctic will undergo the most rapid projected climate change of any other region around the globe (Ford et al., 2021).

## 3 Methodological context and strategy

Hydro-morphological hazards at mid to low latitudes are regularly mapped and their information freely shared in local and global databases. This is the case for co-seismic (Schmitt et al., 2017; Tanyaş et al., 2017) and rainfall-induced (Kirschbaum et al., 2009; Emberson et al., 2022) landslides and the same is also valid for floods (Adhikari et al., 2010). The part of the geoscientific community working on cryospheric hazards has not yet produced global products, but current trends have seen an increase in data sharing, with thaw slumps inventories often becoming part of supplementary materials in recent publications (Ramage et al., 2017; Lewkowicz and Way, 2019; Swanson, 2021; Nitze et al., 2018). Our aim here is to align with this movement and share the latest version of our TS and TEG inventories mapped for the Nordenskiöld sector in Svalbard. In Section 3.1, we provide a detailed description of the two inventories.

Moreover, another aspect differentiates research carried out at mid to low latitudes with respect to the trends in the Arctic context. In fact, hazard inventories have been commonly used for susceptibility modelling since the early years in 1970 (Brabb et al., 1972) and their results are presented both for explanatory (Lombardo and Mai, 2018) and predictive (Lima et al., 2021) purposes. The explanatory element of these models is usually meant to interpret why they occur where they occur based on statistical relations between the locations where these hazards take place and their landscape/environmental characteristics (Steger et al., 2021). As for the predictive aspect of these models, they are used to probabilistically define areas where these processes may currently be absent but their characteristics imply that they could manifest in the future





(Reichenbach et al., 2018). As a result, decision makers can plan suitable remedial actions, if needed, or assign land use
development constraints (Roccati et al., 2021). High-Arctic environments have not received the same modelling attention
with few exceptions e.g., (Blais-Stevens et al., 2015; Luoto and Hjort, 2005), despite their inarguably unique and pristine
vulnerable landscapes threatened by global warming. Therefore, our intent is to expand the available literature on data-driven
models applied to cryospheric hazards and demonstrate their potential as tools to understand local dynamics as well as
predicting locations that will undergo the same surface deformation process.

## 3.1 Cryospheric hazards inventory

To build a comprehensive inventory of the two cryospheric hazards (TS and TEG), the most recent orthophotos (5 x 5 m
pixel size) acquired in 2009-2011 from the Web Map Services (WMS) of the Norwegian Polar Institute (Npi, 2022) were
interpreted. Unfortunately, no subsequent imagery has been collected in the last ten years and the available scenes in Google
Earth and Esri Wayback Imagery are quite coarse and unsuitable for detailed mapping of the relevant features. Most of both
process TS and TEG appear to be fresh or partly active landforms and can thus be considered relatively recent. TS (Fig. 2a)
and TEG (Fig. 2d) were morphologically identified, digitised on-screen as polygons and then quality checked through
extensive field campaigns distributed over three years (2019-2021). During these visits, aerial surveys were also undertaken
using unmanned aerial vehicles (UAV), whose example images are shown in Figures 2b and 2c. In addition to those, direct
photos were also collected (see Figs. 2e and 2f). To complement the field surveys, we also brought a Trimble S5Series
Motorized total station and a Trimble TSC3 controller for long-term monitoring, whose use though was limited to few
specific TEG locations.

The use of deep learning architectures has started to produce interesting results for automated cryospheric hazard mapping,
with viable examples both for TS (Xia et al., 2021; Huang et al., 2020; Huang et al., 2022) and TEG (Huang et al., 2017).
However, their implementation has not matured yet into operational mapping tools and for this reason, we have opted to
manually interpret and digitize the two inventories, with the aim of producing them with highest quality and completeness.

We examined the frequency-area distributions of both TS and TEG inventories based on approaches widely used in the
landslide literature (Malamud et al., 2004; Tanyaş et al., 2018). A few studies show that a power-law exists for medium and
large landslides and the slope of the power-law (power-law exponent) is used to explore a link between power-law exponent
and regional differences in structural geology, morphology, hydrology, and climate (Densmore et al., 1998; Li et al., 2011;
Hergarten, 2012). However, these kinds of analyses are not common for TS or TEG in general. In fact, even the validity of
power-law has not been examined in detail yet. Given this motivation, we analysed frequency-area distribution curves of the
inventories and assigned a fit to each using double-Pareto Simplified function (Rossi et al., 2012). We also checked the
validity of power-law fitting using the Kolmogorov–Smirnov (KS) statistic that generates a p-value indicating the
plausibility of the hypothesis (Clauset et al., 2009). A p-value close to one indicates a good fit to the power-law distribution,
whereas p-value equal or less than 0.1 might indicate that the power-law is not a plausible fit to the data.

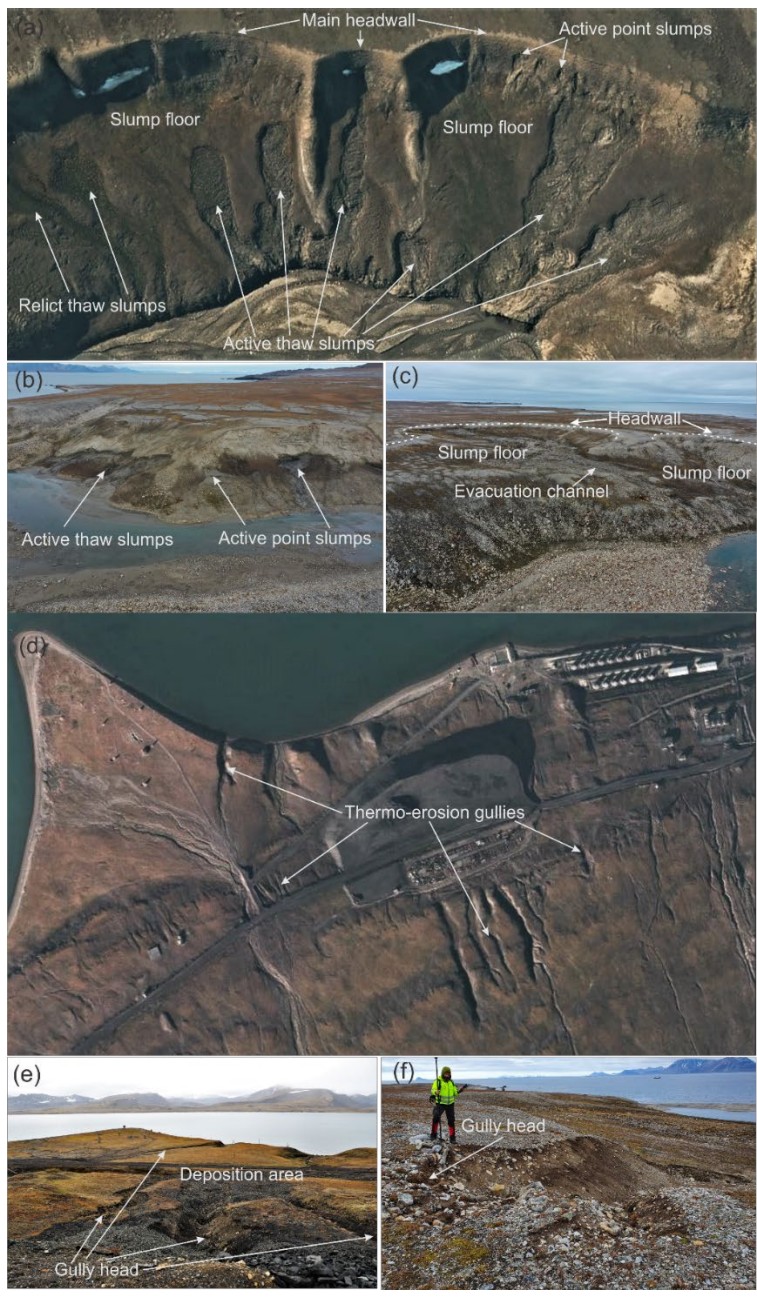

**Figure 2. (a) TS on a north facing slope on the left side of Hanaskogelva River, in proximity of Advent City (orthophoto by © (Npi, 2022) (b) UAV photo of active TS and active point slumps along the right side of Linnéelva river, close to Russekeila. (c) UAV photo of an inactive TS (left side) and an active TS (right side) along the left side of Linnéelva river, close to Russekeila. (d) Thermo-erosion gullies on a western facing slope in Finneset, south of Barentsburg (orthophoto by © (Npi, 2022) (e) Photos of gully heads and their deposition areas south of Barentsburg. (f) Gully head cut into uplifted beach and marine deposits on the left side of Linnéelva river, close to Russekeila.**




### 3.2 Environmental variables for statistical analysis

Due to the terrain settings of Nordenskiöld as well as known morphological and geological attributes associated to
thermokarst activity, and specifically to TS and TEG, we selected several environmental variables (Ward Jones et al., 2019;
Lacelle et al., 2010), which are presented in Table 1.

**Table 1. Environmental variables used in the study**

| Environmental variable | Shortcut | Reference | Unit |
|---|---|---|---|
| Distance to Channel | D2C | (Rudy et al., 2017) | m |
| Elevation | ELV | (Rudy et al., 2017) | m |
| Planar Curvature | PLC | (Nicu et al., 2021a) | 1/m |
| Profile Curvature | PRC | (Nicu et al., 2021a) | 1/m |
| Slope | SLP | (Rudy et al., 2017) | degrees |
| Topographic Position Index | TPI | (Rudy et al., 2017) | unitless |
| Topographic Roughness Index | TRI | (Nicu et al., 2022c) | unitless |
| Topographic Wetness Index | TWI | (Rudy et al., 2017) | unitless |
| Aspect | ASP | (Ward Jones et al., 2019) | degrees |
| Geology | GEO | (Myhre, 2022; Rudy et al., 2016) | unitless |

Out of these covariates the terrain ones originated from a 5 m DEM (Melvær et al., 2014). However, keeping this resolution
would have led to $122 \times 10^6$ grid cells for the whole study area and therefore, we opted to upscale the grid resolution to 100 m
for computational reasons. Also, the Norwegian regulations require that cultural heritage site should be marked under risk, if
closer than 100 m from the nearest cryospheric hazard. Thefore, a grid cell size of 100 m both ensured a reasonable
computational burden for the analyses to be carried our later, and it also represented a meaningful mapping unit for disaster
risk reduction practices. Such operation resulted in partitioning Nordenskiöld Land into ~300 thousands grid-cells. These
have been assigned with a value of the corresponding covariate by taking the mean value of the 5 m. As for the ASP, we
reclassified it into 16 classes, each one 22.5 degrees apart. Then, also for the GEO, we assigned to the 100 m grid cell the
predominant categorical class.

### 3.3 Model training and validation

Our modelling strategy relies on a Generalised Additive Model (Titti et al., 2021). This class of models ensures the same
level of interpretability of the simpler and more common Generalised Additive Model (Atkinson et al., 1998; Titti et al.,
2022) while providing much higher performance, close to more complex architectures belonging to machine/deep learning
(Aguilera et al., 2022). GAM can be used to explain data distributed in a few exponential family distributions (Gamma,



Gaussian, etc.). Among these, the ideal framework to model dichotomous data corresponds to the binomial case, where in

the context of our work, TS and TEG are separately assumed to occur spatially according to a Bernoulli distribution (Bryce et al., 2022). A binomial GAM can be denoted as follows:

$$\eta(\pi) \ = \ \log\left(\frac{\pi}{(1-\pi)}\right) = \beta_0 + f_1 x_1 + f_2 x_2 + \cdots + f_n x_n \ ,$$

where $\eta$ is the logit function, $\pi$ is the probability that the response is present at a given location, $\beta_0$ is the global intercept and $f_n$ are the nonlinear functions estimated for each covariate in the model. In traditional regression problems, the input is a continuous quantity and the output is the same. In our case, the input data for the response variable consists of a vector of zero and ones, standing for absence and presence locations. Conversely, the output is expressed in a continuous spectrum of values that represent the probability of occurrence of our response. Therefore, a series of metrics have been developed

through time to express the performance of binary classifier. All of these can be clustered into cut-off dependent and independent metrics, where the former boils down to the selection of a specific value to reclassify the probability spectrum into a binary dataset, from which confusion matrices can be computed (Bertolini, 2021). The latter type relies instead on many probability thresholds to compute True Positives and Negatives as well as False Positives and Negatives, from which metrics such as Receiver Operating Characteristic (ROC) and their Area under the curve (AUC) can be computed (Hajian-

Tilaki, 2013).

Aside from the context provided above a distinction must be made between binary classifications oriented towards explanatory and predictive assessments. The former interprets the functional relations estimated multivariately regressing the vector of presence/absence with respect to the covariate set. This can be usually done based on the full available information. For instance, in our work this implies using 100% of the grid cells of our study area. However, the estimated results cannot

be interpreted for prediction, and this is achieved via two common approaches. If temporal data are available, then the prediction skill of a given classifier can be measured by matching the susceptibility estimated from a given time over the presence/absence distribution of the subsequent period. However, this is a rarely performed task because multi-temporal hazard inventories are still not common (Guzzetti et al., 2012). This is even more valid in peri-Arctic environments, where hazard inventories are scarce even in their pure spatial form. Therefore, when the data dimension is spatially confined, a

well-established routine to estimate predictive performance relies on splitting the spatial data into a portion used for calibration and another one for validation, under the assumption that spatial replicates mimic the behaviour of temporal ones. The training and test split though, can also be done in diverse ways. The simplest corresponds to a pure random cross validation (RCV; (Roberts et al., 2017), although such practice usually leaves the data structure like the original set, therefore also returning similar performances to the calibration ones. A complementary validation routine uses a spatially

constrained subset of the data instead. This is usually referred to as spatial cross validation (SCV; (Brenning, 2012) and offers the ability to assess sectors of a given study area for which the model may locally perform well or fail.

In this work, we make use of all the elements described above: we fit the presence absence data to the whole Nordenskiöld landscape and we use the results for interpretation. As for assessing the predictive skill, we also perform the two cross-validation (a tenfold RCV and an eightfold SCV), for both TS and TEG.

## 4 Results and discussion

The resulting inventories encompass 562 TS and 908 TEG. Compared to the previous preliminary study (Nicu et al., 2021a), the RTS inventory has increased from 400 polygons to 562 polygons. As for the TEG the updated version of our inventory included 908 polygons, 98 more than what was mapped in a previous study (Nicu et al., 2022c). This final effort brought our inventories to their current and final form, where the mapping procedure covered the whole study area shown in Figure 1 and field surveys have validated some of their positions and extent.

The inventories are of high value in a climate change context, as they can be of use by a wide range of scientists, such as geomorphologists, climatologists, hydrologists, biologists, archaeologists, as well as stakeholders and local authorities, in their effort to quantify the potential impacts of the two hazards on infrastructure (Hjort et al., 2018; Hjort et al., 2022) and cultural heritage (Nicu et al., 2021a; Nicu et al., 2022c). To explore their characteristics for any of the users and uses mentioned above, below we will summarize the Frequency Area Distributions (FAD) of the two inventories we mapped and in the subsequent sections we will present the results of the susceptibility modelling we performed.

### 4.1. TS and TEG size characteristics

First, we checked the validity of the power-law for the generated dataset. Based on the KS test, we calculated p-values, which are larger than 0.1 for both TS (p-value=0.6) and TEG (p-value=0.4) inventories. This shows that for both inventories, double-Pareto Simplified function is a numerically plausible fit to the data (Fig. 3). Second, we identified power-law exponents. Power-law exponents simply show the ratio between small and medium/large landslides. In our case, we are calculated them as 2.41 and 2.48 for TS and TEG, respectively. Interestingly, these values gave a perfect match with observations carried out for landslides triggered by earthquake, rainfall, and snowmelt where average power-law exponent centralized around 2.4 and 2.4 (e.g., Malamud et al., 2004, Tanyas et al., 2018). Among numerous factors controlling the power-law exponent of landslide inventories, topography is one of the most mentioned parameters in the literature (Ten Brink et al., 2009). Here, our results show a clear match between power-law exponents of landslides and TS/TEG, although TS and TEG are not generated along steep hillslopes as landslides do. Examining the reason behind this similarity is beyond the scope of this contribution. However, our results indicate that more TS and TEG inventories need to be generated to better understand their size statistics and factors governing the shape of their frequency-area distributions.

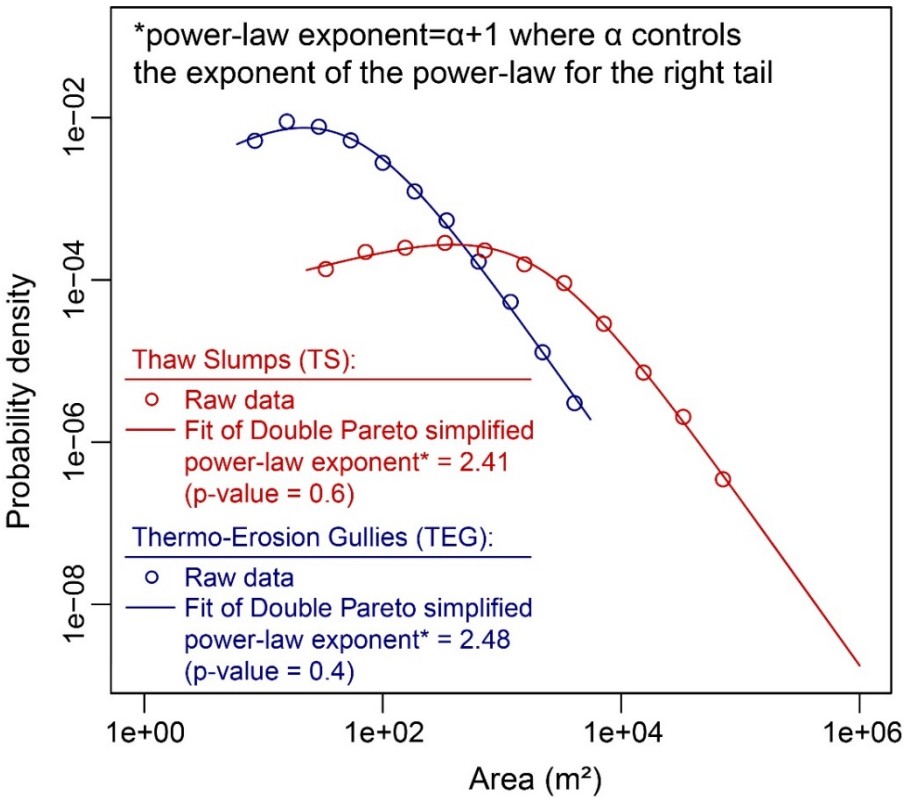

**Figure 3. The FAD obtained for the two inventories in Nordenskiöld Land.**

## 4.2. Susceptibility modelling performance

We measured both goodness-of-fit and predictive skill of our modelling framework. Figure 4 reports the corresponding ROC and AUC values, for the reference fitting procedure as well as the two cross-validations. For both cryospheric hazards, the
280 performance falls within the excellent category according to the AUC classification proposed by (Hosmer and Lemeshow, 2000). At a closer look though, the fit and RCV almost fall within the outstanding class (all the means are above 0.8 and below 0.9). The performance loss exhibited for the SCV is to be expected and it represents an important indication. In fact, it highlights the prediction skill of our model assuming it to be blind to the characteristics of specific portions of the study area. Therefore, a spatial cross validation can be interpreted as the worst situation one can examine to understand a model
285 prediction. Another element worth of being stressed it that the variability for the RCV is clearly low since a random selection is not able to disentangle local spatial dependence in the data. As for the SCV, where the spatial dependence is perturbed due to the constrained local selection, the variability is still within an acceptable range.

Earth System
Science
Data

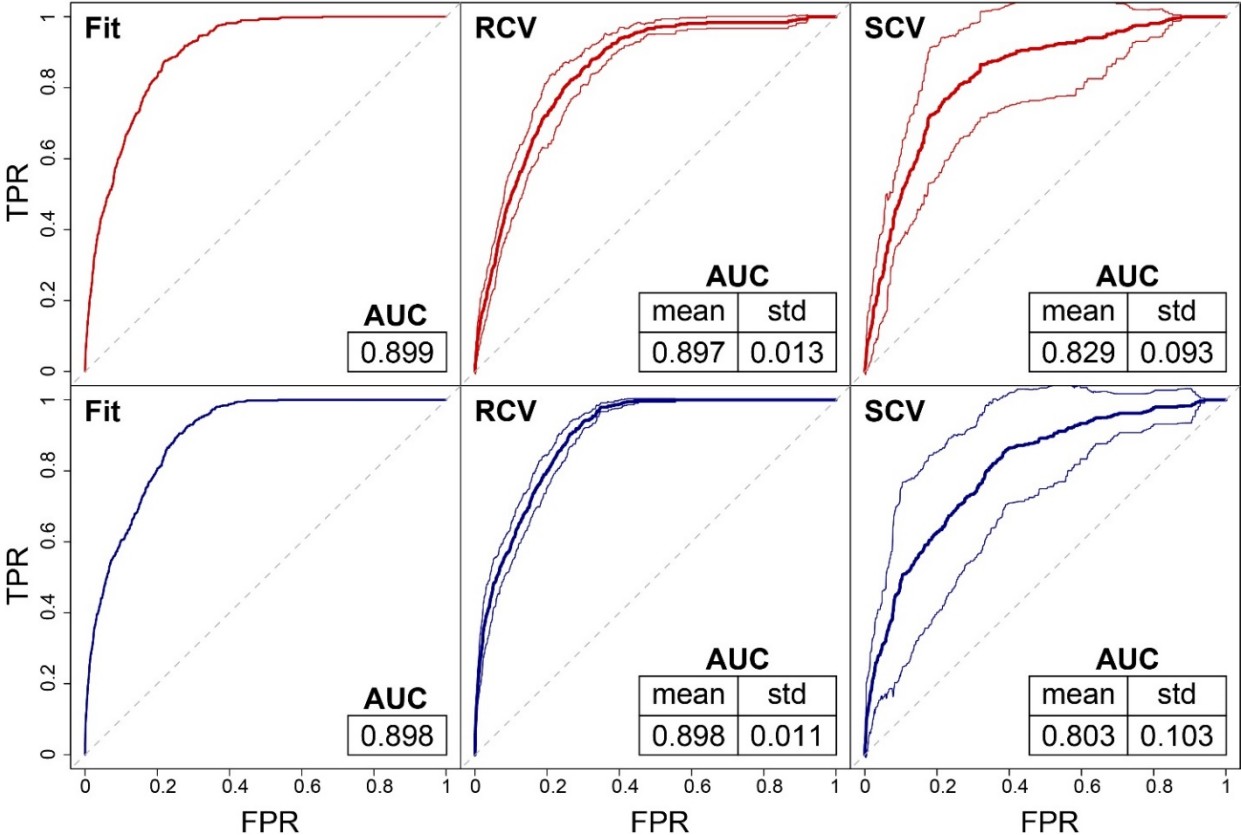

**Figure 4. Modelling performance overview. First row indicates the results for TS whereas the second row reports the TEG. The thick lines for the two cross-validation schemes represent the mean ROC curve, whereas the thin lines graphically summarize the variability of the cross-validation scheme via a single standard deviation.**

### 4.3. Controlling factors of TS and TEG

From the original list of covariates shown in Table 1, we removed TRI and TPI because of a variable selection procedure. Specifically, their inclusion was slight lowering the model performance and inflating the uncertainty in the other nonlinear covariate effects, both for TS and TEG. At a closer look, we noticed that TRI was linearly related to SLP with a Pearson's correlation coefficient ($\rho$) above 0.9 whereas TPI showed a close dependence with respect to PLC attested by a $\rho \sim 0.8$. Figures 5 and 6 provide an overview of the selected covariate effects we used to model TS and TEG respectively. The most striking element of the two figures is that the two processes we modelled share some similarities in the way some of the covariates are influencing their occurrence, although some marked differences also exist. For instance, both TS and TEG occupy the lowlands of the Nordenskiöld landscape, being the ELV contribution dominant within the first 200 m above sea level, after which the effect rapidly decays and becomes heavily negative after a height of approximately 300 m. From a first glance this indicates a positive relation of both processes to sorted and fine-grained sediments, which are found as



isostatically uplifted marine and glaciomarine sediments along the coasts and as fine-grained valley-fills of fluvial and aeolian origin in the rest of the landscape of lower elevations. Conversely, it speaks against a connection to the often extensive sediment covers of in situ weathering material on the higher mountain plateaus. This initial consideration about TS/TEG co-existence is enriched when considering other covariates' effects.

Differences start to arise examining the D2S, which strongly contributes to TS occurrences within tens of meters and drastically drops after that, up to negative effects after few hundreds of meters away from the channel. This effect may have to do with riverbank erosion at the base of a potentially unstable permafrost slab, which once it misses its support starts moving, and further develops into a retrogressive slump. It might also be secondarily linked to some snow-bank effects on the initiation of TS, where thermal conductivity through percolating meltwater from the snow during summer seasons might be of importance. The arctic winters with often high wind speeds favour intensive redistribution of snow over the landscape, accumulating in low-positions, for example channels. Interestingly, this is not the same effect shown for the TEG case where the contribution to the susceptibility is shown to increase 500 m away from a streamline. This may be because a gully to form, it needs an incision to develop into. A streamline represents an incision that has already widened in time and therefore, it is only reasonable for TEG to manifest a bit further away.

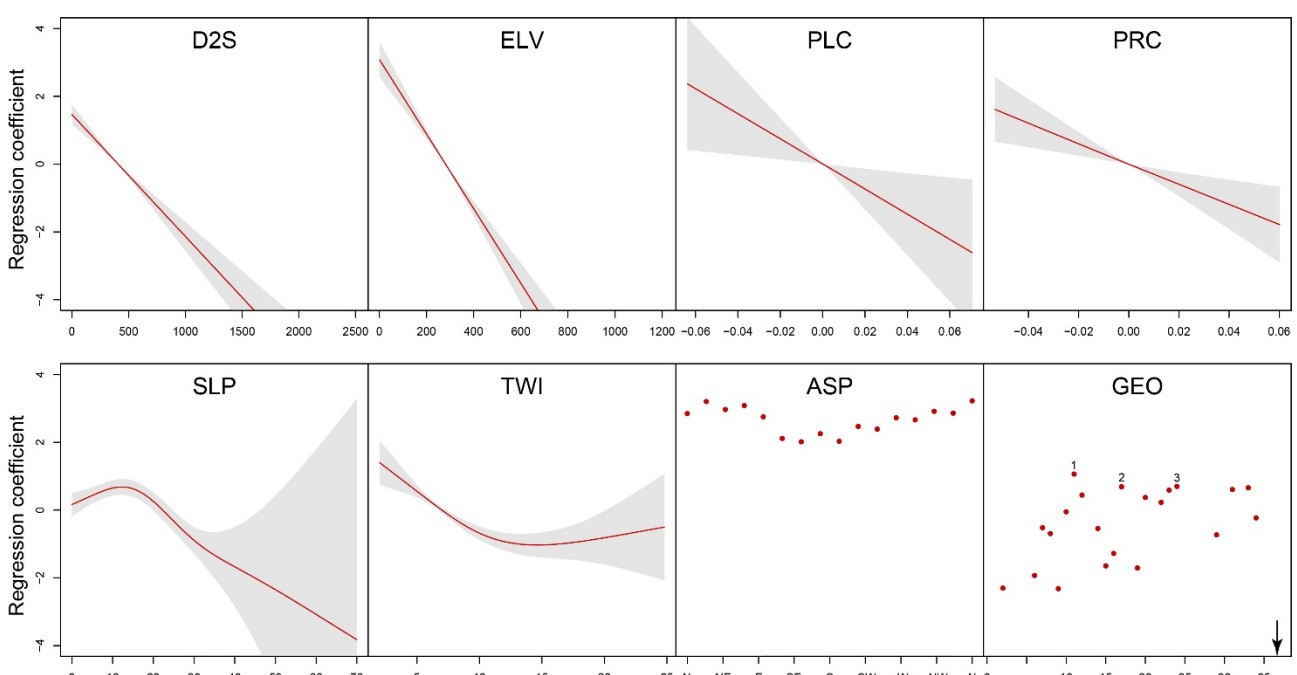

**Figure 5. Covariate effects estimated for TS. Notably, the regression coefficients estimated for outcropping lithologies also host strong negative values. For pure visualization purposes, we have focused on representing a coefficient range where the positive classes would still appear to be visible. Also, to avoid cluttering the text, we have described in the text only the three strongest and positive contributors, labelled 1,2,3 in the image.**



The image of the landscape prone to the two processes can be further diversified looking at SLP, where both processes show a quite different behaviour. The probabilistic occurrence of TS is favoured up to 20 degrees after which, the SLP

contribution becomes increasingly negative. As for TEG, the overall SLP contribution appears negligible, with the first tents of degrees being slightly positive and the remaining steepness domain becoming slightly negative. This indicates once more that TS do form in near flat areas whereas TEG can also occur along steeper morphologies. There, the overland and/or interstitial flows would accelerate over preferential directions giving rise to linear erosion forms that may further develop into gullies. As for the exposition, some degree of similarity can be seen once more, with the North, North-East and North-

West directions contributing to increase the probability of TS and TEG occurrence.

The geological control is extremely complex and would require listing tens of lithotypes; however, this is not the primary focus of this paper. Firstly, we can conclude from field and remote sensing data, that most, if not all both TS and TEG occurrences are situated in soft sediments (Quaternary deposits). This is not surprising, given that they both rely on grain-to-grain conditions with and without permafrost internal ice. Since Nordenskiöld Land lacks continuous data of Quaternary

geological sediment, this is not included in the statistical analysis. Knowledge of the connection between bedrock and deposition of sediments indicates that local bedrock is however often linked to the soft sediment deposits. This assumption is especially true for *in situ* weathering slope deposits, fluvial deposits, and glacial tills, but a little less obvious for marine deposits. This relation prompted us to look at bedrock lithology (where regional data is available) as one factor in the analysis.

For reasons of conciseness, we opted to report the three highest contributors with a positive sign, to express litho-type characteristics prone to host TS and TEG. Specifically, the probability of TS appears to increase in areas overlying bedrock of shales (bituminous), siltstones and sandstone mixed deposits dated back to the Late Jurassic - Early Barremian. This is again the case for bituminous shales and siltstones mixed deposits originated during the Late Jurassic. And the third lithotype prone to TS is also the highest contributor for TEG, this consisting of shales, mudstones, and siltstones of the late

Palaeocene. The second highest geological contributor for TEG consists of a mixture of sandstones, shales and coal formed again during the Palaeocene and the third one is represented by a deposit hosting sandstones and conglomerates of the Barremian. This is clearly an interesting description of the geological effects, because the model out of many different classes consistently picked the same lithotypes as predisposing factors for TS and TEG, with minor differences represented by coal and conglomerate inclusions.

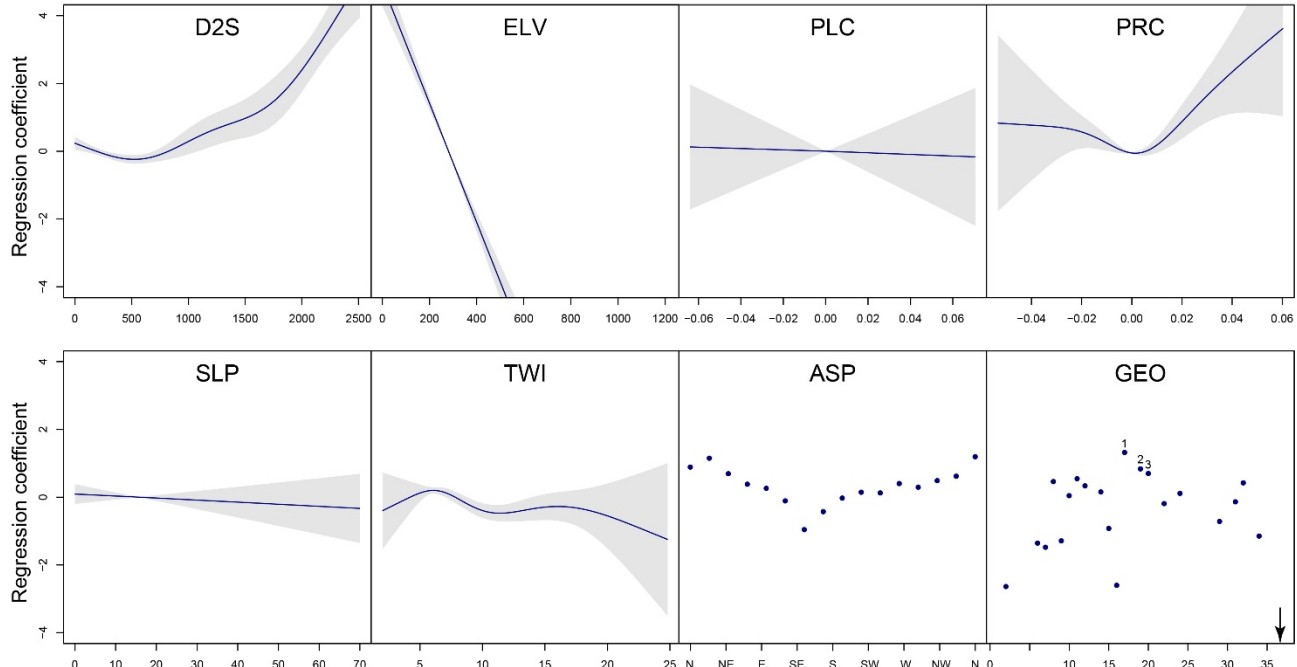

**Figure 6. Covariate effects estimated for TEG. Notably, the regression coefficients estimated for outcropping lithologies also host strong negative values. For pure visualization purposes, we have focused on representing a coefficient range where the positive classes would still appear to be visible. Also, to avoid cluttering the text, we have described in the text only the three strongest and positive contributors, labelled 1,2,3 in the image.**

## 4.3. Susceptibility mapping of TS and TEG

The good performance and the reasonable effects presented above suggest that the models we produced for TS and TEG are reliable and can be considered for susceptibility mapping. To graphically summarize this task, we produced two overviews, one where the susceptibility values are shown in their continuous form and one where we grouped them into classes. Figure 7 returns these two options both for TS and TEG.

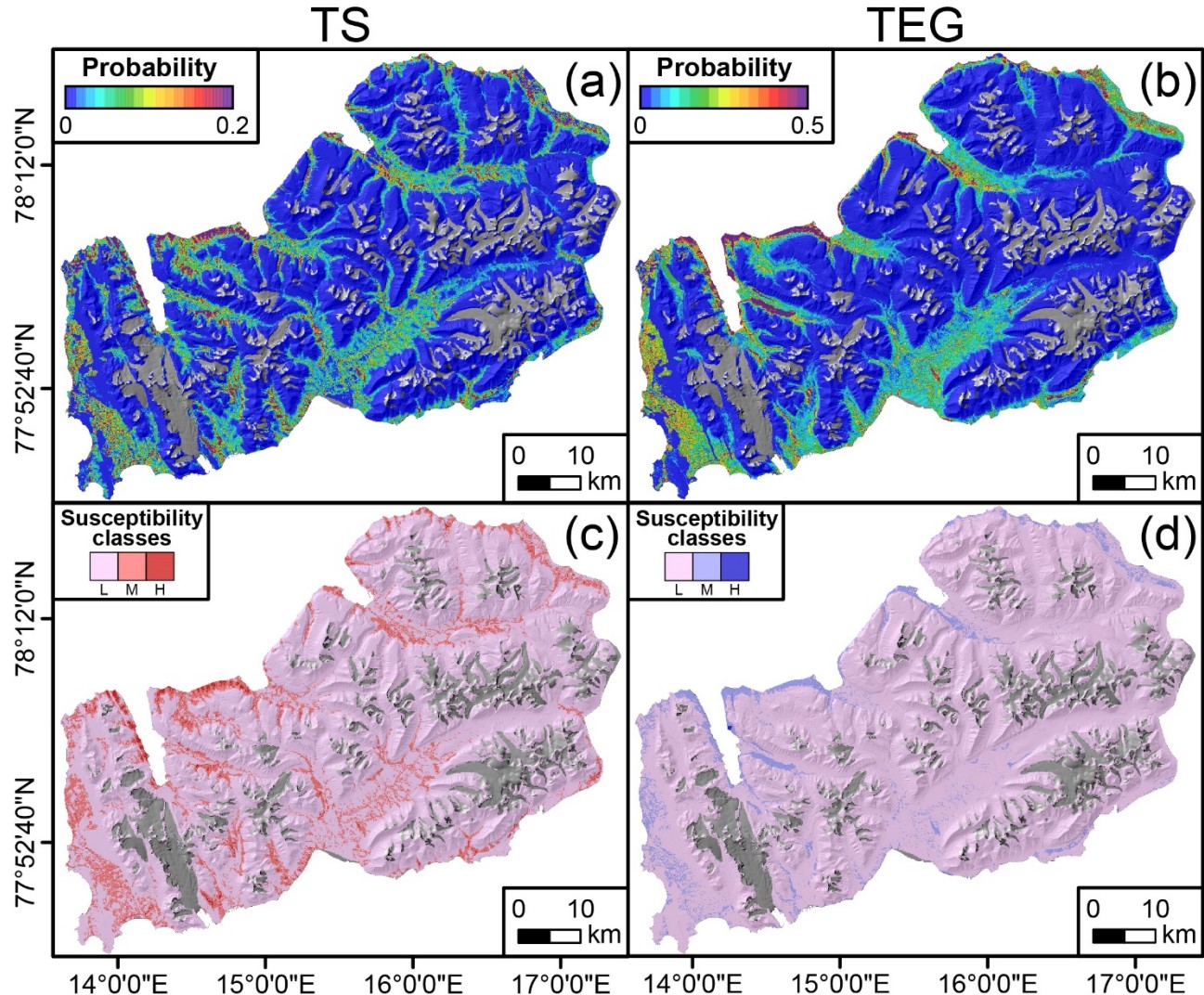

**Figure 7. Susceptibility map reporting the probability in its continuous (a,b) and classified (c,d) forms, for TS (first column) and TEG (second column), respectively. Grey areas correspond to glaciers, which have been masked out from the analyses. The classification followed the Jenks methods, by minimizing the within-class variance after an arbitrary choice of three classes (L for low, M for medium and H for high susceptibility).**


The TS susceptibility patterns (left column) appears to be distributed along the coastlines and in part of the central-western sector of Nordenskiöld Land, supporting the link to the raised marine deposits. Specifically, coastal areas likely to host new formations of TS can be found between Heerodden and Eriksonodden (Colesbukta), Festningsodden and Kokerineset (western part of Grønfjorden), scattered areas between Kapp Linné and Kapp Starostin, Vestpynten and Adventpynten (close

to Longyearbyen Airport), and in the northern part between Diabasodden and Elveneset, Vindodden. The reason these





locations are relevant for the Nordenskiöld community is that they are also locations where or close to where most human activities take place on the island. As for areas susceptible to TEG (right column), these are characterised by higher probability of occurrence while also being more concentrated in few areas. These areas overlap with main human settlements (Longyearbyen and Barentsburg) and former mining settlements (Grumant and Svea), to the point that it raises the question

whether the formation of TEG may be partially due to anthropic effects. Other than being a speculation though, no obvious signs of such spatial dependence were found during our fieldwork activities and thus it is an observation we opted to share with the readers but also to reject from our own experience.

It is worth mentioning that the difference in probability range shown for the two cryospheric hazards is also because TEG are more numerous than TS, thus the different proportion of presence and absence data influences the global intercept, making it

less negative for the TEG than for the TS. However, this effect still allows for the spatial predictive patterns to be suitably depicted, with differences that emerge based on the landscape characteristics. Nevertheless, these patterns are still portrayed in a separate manner, therefore making it difficult to perceive areas where they clearly co-exist. In the next Section, we will address this issue by providing details on how we generated a map capable of showing the probabilistic assessment of multi-cryospheric hazard occurrences for the Nordenskiöld Land.

**4.4. Multi-cryospheric hazard susceptibility mapping**

To simultaneously represent the likelihood of TS and TEG within the same map, we opted to combine the two reclassified maps previously shown in Figure 7. The resulting multi-hazard susceptibility map is shown in Fig. 8, where nine classes are portrayed through a two-dimensional colour bar, reflecting the RGB (red-green-blue) combination of the three classes per hazard in Figure 7. Most of Nordenskiöld falls in the LL category and the extent of the other eight classes exponentially

decreases as the combined susceptibility level increases. However, being the site extremely large, this still implies that quite some portions of the territory may be subjected to either or both cryospheric hazards. For this reason, we also report the total extent of the nine classes (whose graphical expression is plotted as a bar plot within Fig. 8, with LL covering 2657 km$^2$, LM 244 km$^2$, LH 4 km$^2$, ML 37 km$^2$, HL 0.48 km$^2$, MM 112 km$^2$, MH 20 km$^2$, HM 0.04 km$^2$ and HH 0.03 km$^2$.

The most susceptible areas to both cryospheric hazards are located along the coastlines from the central, north western,

south-western, and north-eastern parts of Nordenskiöld Land. Three examples are also shown in Figure 8 to highlight details of the multi-hazard estimates. These are locations of actual relevance for the Nordenskiöld Land, as it is prone to human settlements and poses danger especially to infrastructure. Z1 shows the area prone to the Stemmevatnet lake, which represents the main water resource for Barentsburg. Any future TS and/or TEG processes may jeopardise this aspect. Z2 highlights the area north of Barentsburg, where important infrastructure is located. And finally, Z3 shows the main

settlement, Longyearbyen, along with the area around the airport. This is of high importance for local authorities and stakeholders in their effort to minimise future disturbance of the local infrastructure.

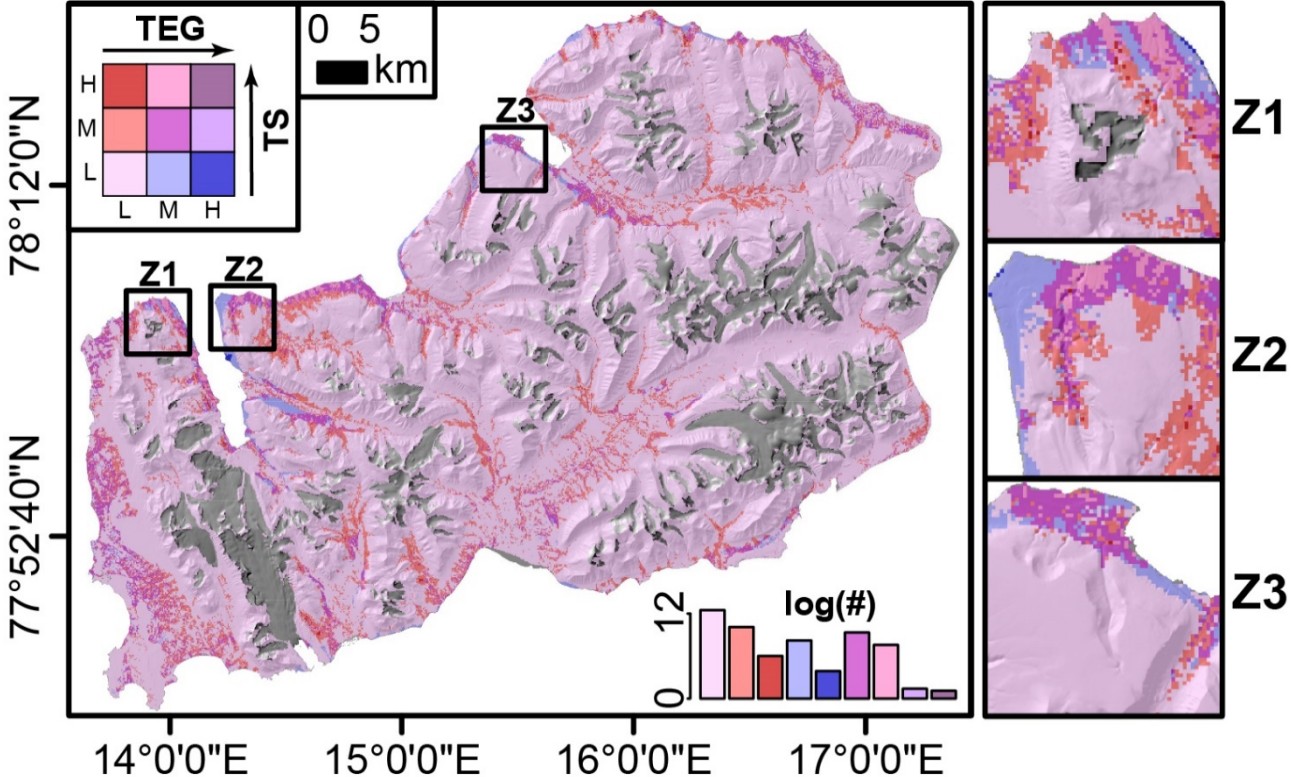

**Figure 8. Multi-hazard susceptibility map of TS and TEG for Nordenskiöld Land. Zooms 1,2 and 3 highlight portions of the territory where the two cryospheric hazards can interact with human activities and local infrastructure. The bar plot at the bottom right represents the number of grid cells expressed in logarithmic scale, for each of the nine combined susceptibility classes.**

## 5. Considerations within and beyond Svalbard: supporting and opposing arguments

A systematic TS and TEG mapping protocol to share these cryospheric hazards among researcher has yet to root within the geoscientific community. This work aligns well with other attempts to make data on TS and TEG become freely accessible, because we believe that the surface deformation dynamics of delicate environments laying within Arctic and peri-Arctic regions can be studied only as a collective effort. For this reason, we share our inventories, in the hope of triggering similar behaviours within our community and to stimulate the implementation of advanced models, as per other mid-latitude hydro-morphological processes. Aside from the importance of a standard data sharing platform within the global system, even just within the Svalbard context this is something of great relevance. In fact, the study site we chose had undergone significant changes in recent times. The work of (Ziaja, 2001) has shown the extent of these changes in the form of permafrost degradation, whose dynamics can be better understood if framed within the bigger picture of the Svalbard meteorological settings. In fact, Nordenskiold Land has always been covered with a lesser glacier extent compared to the rest of the



archipelago. This is due to the direction the maritime air masses follow in the area. Specifically, the effect of the warm West

Spitsbergen Sea Current creates a convergence of mild and humid air from the South and cold air from the North. This convergence results in a local micro-climate warmer than the rest of Svalbard and in general than what is typical at these latitudes. In addition to an already delicate situation, Ziaja (2001) observed that the deglaciation in Nordenskiold Land has evolved at a double rate compared to Sorkapp Land (south Svalbard), arguing this to be an indication of a greater sensitivity of our study site to global warming. Therefore, we consider vital documenting and sharing evidence of permafrost

degradation (our TS and TEG inventories) to reconstruct a baseline to which future monitoring protocols should refer to, to further explore the effects of climate change in the area. One of the possible tools to use to explore these effects falls in the category of data-driven models, among which susceptibility study are part. However, hardly any susceptibility studies have been carried out so far to estimate locations prone to TS and TEG in peri-Artic regions (Blais-Stevens et al., 2015; Rudy et al., 2016; Veh, 2015). Along this line of research, we proposed a tool for interpretable and flexible predictive models,

offering the chance to explore the results from multiple aspects, among we include a multi-hazard susceptibility assessment. The performance it produced falls within the excellent class proposed by (Hosmer and Lemeshow, 2000). Therefore, standard practices would consider such model results a reliable information for local administrators to base their decisions and plan suitable course of actions to reduce the risk due to these cryospheric hazards.

This is already an important achievement, however below we would like to stress a few elements that we already envision

requiring further considerations, to develop our model into an operational tool.

Both TS and TEG processes are shown to be highly dependent on soft sediment characteristics, data which so far lacks on Svalbard. Adding map-data with type and potential thickness of surface sediments would further increase the accuracy and detail of predictions. The other, even more prominent issue we faced had to do with the absent temporal information of our inventory. This is something that unfortunately affects virtually all the TS and TEG inventories mapped across the globe. For

this reason, we are limited to statically investigate and understand locations prone to these hazards. However, this also raises the question whether such information can be really used outside the academic context. In fact, any model without a temporal connotation will inevitably learn to mimic the process occurred at the time of the orthophoto or satellite image used for mapping. In other words, no temporal information on temperature, rainfall and other dynamic characteristics can be included in the model. Therefore, in a rapidly changing environment such as the Svalbard landscape, the probabilities of

occurrences we estimated may have already been affected by global warming and permafrost degradation processes (Ziaja, 2004). With this in mind, we consider our workflow just a proof-of-concept of what can be achieved, in the hope that the years to come and a broader scientific effort can bring together a fully spatio-temporal description of these cryospheric hazards. If this wish would become a reality, then a whole spectrum of different models and research questions will open for the geoscientific community to address. For instance, future simulations of TS and TEG probabilities at varying climate

scenarios could be achieved by introducing, for instance, the temperature as a covariate and then using a plug-in simulation (Do et al., 2005; Lombardo and Tanyas, 2020) tool to project the change in susceptibility as the future temperature pattern changes. Fortunately, the current status of the scientific branch focused on developing automated mapping tools has reached



a level of maturity close to become; widely adopted even in peri-glacial environments (Meena et al., 2022; Nava et al., 2022). For instance, a first article has already been published on the use of deep learning architectures for automated TS

mapping (Huang et al., 2020). This represents a promising venue for multi-temporal mapping because each artificially intelligent mapper tool is run over a specific remotely sensed scene and the same operation can therefore be repeated for each satellite orbit. Still remaining is the lack of spatially detailed and accurate data from the Arctic, where the processes discussed here required a cca. 5 m resolution for accurate detection of features to form a training dataset.

Another element that can be improved with future efforts has to do with the actual target of the model. So far, our aim was to

estimate locations prone to TS and TEG formation. However, these processes have also a spatial extent and the threat they may pose on local activities is equally if not more important than the simple notion of where they may initiate. For this reason, we already envision future models that would take the measured extent of TS and TEG as the response variable, this time solving a regression task rather than a classification, one as per susceptibility requirement. Such direction has recently been explored for landslides occurring at lower latitudes (Lombardo et al., 2021; Moreno et al., 2022). And, an even better

extension has already been tested where the expectation of locations prone to landslides are modelled together with the expectation of the resulting landslide size (Aguilera et al., 2022; Bryce et al., 2022).

Notably, all these methodological considerations are valid extensions to be tested within the Svalbard landscape. However, they can also be valid outside it. If space-time models would become a viable approach because multi-temporal inventories would also become available, then dynamic simulations could also be extended to the whole peri-arctic sector. This would

enable large scale considerations on climate change and its cascading influence from temperature to TS and TEG spatio-temporal patterns.

At a global level, permafrost is undergoing considerable degradation following the increasing trend of lobal warming. The most recent IPCC assessment reported with high confidence that the Arctic warmed at more than twice the global rate over the past 50 years with the greatest warming during the cold season (Constable et al., 2022). This leads to TS and TEG

occurrences, which can put a threat to Arctic infrastructure (Hjort et al., 2022), cultural heritage (Nicu et al., 2021a; Nicu et al., 2022c), impact the fluvial sediment budget (Lamoureux and Lafrenière, 2018), release significant amounts of greenhouse gases, such as carbon dioxide and methane to the atmosphere (Oberle et al., 2019; Ran et al., 2022). Cryospheric hazards are likely to further increase in the future following climate change (Ding et al., 2021), and using the latest statistical advances to predict their likely occurrences is of paramount importance. This study showed the importance of the two inventories and

what can be achieved when using them both separate and together in a multi-hazard approach. The method can be adapted and transferred to the entire ice-free area of Svalbard Archipelago and other circumpolar areas. The final multi-hazard map represents a valuable tool, that can be further processed and improved, for local authorities and policy makers, and can be transformed into plans at various scales of mitigation measures.



## 6 Conclusions

At a global level, permafrost is undergoing considerable degradation following the increasing trend of global warming. The most recent IPCC assessment reported an Arctic warming at more than twice the global rate over the past 50 years, with the greatest warming during the cold season. To better understand what are the expectations of future permafrost degradation-related processes, a systematic sharing practice of the mapping routines we perform as a community should become commonplace. In line with this objective, in this work, we share the TS and TEG inventories we mapped and validated 490 through several field campaigns. Moreover, to better understand these processes and attempt to reliably predict them, the implementation of data-driven models holds a promising potential. This is also the case for cryospheric hazards such as TS and TEG, whose occurrence probability we propose here to be modelled via a binomial GAM. We also take a step further and produce a multi-hazard susceptibility map of our test site in Nordenskiöld Land. These types of models are also rare in peri-Arctic environments and their spread may lay the foundations to build a global assessment of cryospheric hazards' 495 development as a function of global warming. This is the direction we consider to be crucial to assess the risk that Arctic communities may soon be exposed. This is something of fundamental importance because the changes we have witnessed in the recent past and that we see today will be relatable to the changes we will see in other permafrost-rich areas such as the Alps or the Himalayan range. Their global warming is yet to reach the extent of the change we have observed so far near the pole and therefore in Svalbard.

**Data availability**

The NPI images are freely available at https://toposvalbard.npolar.no/. The Digital Elevation Model is freely available at https://data.npolar.no/dataset/dce53a47-c726-4845-85c3-a65b46fe2fea. The Geological Map of Svalbard (Geologi Svalbard), in raster format, scale 1:250 000 is freely available at https://geodata.npolar.no/arcgis/rest/services/Temadata/G_Geologi_Svalbard_Raster/MapServer. The TS and TEG 505 inventories are publicly available in shapefile format at https://doi.org/10.1594/PANGAEA.945348 (Nicu et al., 2022a) and https://doi.pangaea.de/10.1594/PANGAEA.945395 (Nicu et al., 2022b), respectively.

**Author contributions**

ICN and LL designed the study. ICN prepared the initial datasets and wrote the draft. LE, HT, and LL designed the methodology and performed the statistical analysis. LR validated the initial datasets and contributed to the draft. ICN, LE, 510 LR, HT, and LL improved the writing and structure of the final manuscript. All authors agreed on the final version of the manuscript.



## Competing interests

The authors declare that they have no conflict of interest.

## Acknowledgements

I.C.N. was partially supported by Fram Center Flagship Klimaeffekter på økosystemer, landskap, lokalsamfunn og urfolk, GEOCULT – Monitoring Geohazards Affecting Cultural Heritage Sites at Svalbard [grant nr. 369913 and 369924]. L.L. was partially supported by King Abdullah University of Science and Technology (KAUST) in Thuwal, Saudi Arabia, Grant URF/1/4338-01-01.

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
