# Peer review of "Multi-hazard susceptibility mapping of cryospheric hazards in a high-Arctic environment: Svalbard Archipelago"

_Earth System Science Data, 2022_

## Author Comment (AC1)

Dear Reviewer,

We thank you for your suggestions and observations. Below, we answered point-by-point to your comments. In black (your comments), in red (our comments).

Review – Nicu et al. 2022 „Multi-hazard susceptibility mapping of cryospheric hazards in a high-Arctic environment: Svalbard Archipelago"

The authors present a detailed inventory of thaw slumps (TS) and thermo-erosion gullies (TEG) of a central part of Svalbard archipelago in the High Arctic. As such, the database is rather unique and can serve for multiple purposes ranging from further research of cryosphere related processes to actual mitigation actions of local policy makers. In general, the paper is well written with no important issues that would cause serious problems in using the database.
We highly appreciate your positive general comment on our paper.

My biggest concerns are related to reliability and quality of source data and the necessary generalisation within the modelling process. However, authors included a detailed discussion of the limitations (especially in section 4.3). This includes also the factor of geology/bedrock which, in my opinion, is the principal controlling factor affecting spatial distribution and main parameters of both TS and TEG.
We tried to possibly comment on our data limitation in section 4.3., which we think is a step forward in the context of multi-hazard mapping, especially in an Arctic environment. We agree that the geology in the form of unconsolidated sediment types logically is a major controlling factor for spatial distribution of both TS and TEGs and would happily have included this as a major factor in the modelling. However, there is an almost complete lack of published spatial data over unconsolidated sediments in the study area (as indeed for the whole of Svalbard). This is the same for geomorphological maps. There is no comprehensive mapping program for Svalbard, in difference to the bedrock geology. The Bedrock-Geological data record unconsolidated sediments as one unit, and these overlaps (as is logic largely) with the TS and TAGs. However, one would expect more complex and important relations with different types of unconsolidated sediments (e.g., glacial tills contra fluvial contra raised marine deposits), but as mentioned there is a lack of comprehensive data on the coverage comparable to our extensive morphological inventories.

I have several major comments, which I would like the authors to answer and possibly adjust in the paper.

a/ I understood that the NPI aerial images were used for delimitation of the TS and TEG. Beside this you described that you had several extensive field campaigns including UAV mapping and dGPS measurements – this seems not to be used any further in the process. Can you comment on that?
Thank you for this observation. Indeed, the NPI aerial images were used to delineate the TS and TEG. The field campaigns using UAV and a total station (not dGPS) were very limited in time. The total station measurements were rather used to monitor small-scale gullies, and not the big ones that we refer in this study. However, the extent of those gullies was also included in this inventory. Moreover, the field campaigns were rather used to familiarize with the morphological "fingerprint" of the TS and TEG in the landscape, so that we would do a better identification on the aerial images; and to make very suggestive field photos (as ones in Figure 2).

b/ I miss any information on the minimum length/area of the features that are included in the database – I guess this is related to the spatial resolution of the aerial images.

An extra column was added in the attribute tables of both TS and TEG referring to area (in m$^2$).

c/ It might be good to include some measure of uncertainty in the delimitation process – this is related to the resolution as mentioned above.

The uncertainty can be hypothetically addressed by the fact that we do mention the resolution of the aerial images, which is 5 x 5 m/pixel. Therefore, there should be no gully that is below, as an area, this threshold.

d/ I wonder if some estimate of volume of the mapped features could be included in the database. I think that the area of TEG is interesting, but the volume of transported material might be even more beneficial for example for policy makers when planning mitigation actions in the settlements.

I was actually expecting that the UAV flights were intended for volume estimation. This might be also done using 2m resolution ArcticDEM for larger features. I am completely aware that it would need a lot of time to be done for the whole dataset, but at least for a few sites perhaps?

We know that some volume estimation would be interesting, but unfortunately, calculating volumes falls outside the scope(s) of the present paper. The UAV flights were no intended for volume estimation, as they were made on very limited surfaces, to monitor small scale TEG and TS and have a better image on how they look from a better quality aerial image. Volume estimation is one of our future endeavours and we will follow up on this with future papers.

e/ Would it be possible to quantify the importance of controlling factors? In a very simple way with use of PCA or similar statistical approach? That would be a nice outcome of the whole work.

This has been explained and detailed in section 4.3. Again, making such an analysis does not fall in the scope of this paper. More details about the controlling factors can be found in the two previous papers that we published. However – as mentioned in the beginning of this discussion, we are well aware of the importance of the properties of unconsolidated sediments in which both TS and TEGs are developed, and this is one of our plans for future work.

f/ Do you think that the model calibrated on the data from Nordenskioldland could be easily used in other parts of Svalbard or elsewhere? Can you comment on that in the discussion?

We have already mentioned this aspect towards the end of the Section 5.

g/ I would suggest including other derived parameters of the mapped features directly in the database as an attribute of the shapefiles (area, slope, aspect, centreline length, bedrock type and so on). I guess you have these data and others might benefit from that without need to proceed the analyses on their own. In the TS attribute table an "age" column is included with values 1 or 2 – can you explain that?

We have now added in both databases a column with area (in m$^2$), perimeter, P/A, P/sqrt(A), Max. Distan, D/A, D/sqrt(A), Shape index. The TS age-column was only a very primary attempt at distinguishing possible relative ages

based on vegetation cover and morphology, but over the mapping time it became clear that the quality of aerial photographs available were not good enough for certain conclusions on this aspect and we have removed it from the dataset, hoping for future data to corroborate the tested approach (we deleted it).

Technical comments:

a/ please include scale in Fig 1 and 2.

We have now included the scale for Fig. 1 and Fig. 2. Moreover, in Fig. 2, we also included the north direction for more clarity.

b/ please doublecheck the reference list, I wanted to look at Myhre 2022 (in Table 1), but it is missing in the references

This is reference to a webpage – Geologi / Geology of Svalbard. It is from NPI, but it appears in the references as Geologi / Geology of Svalbard. Now has been corrected

c/ I would suggest marking where the settlements are located in Fig 7 and 8 – that might help the readers not familiar with Svalbard.

We thank you for this suggestion, but adding the settlements would just make the map more "crowdy" and "heavy" to understand. The main human settlements are visible in Fig. 1.

d/ it might also help redesigning Fig 8 and enlarge the Z1, Z2, Z3 + insert schematic infrastructure (buildings, roads) to illustrate how does the susceptibility interact with actual existing infrastructure

We have now done that in the new Fig. 9.

---

## Author Comment (AC2)

Dear Reviewer,

We thank you for your suggestions and observations. Below, we answered point-by-point to your comments. In black (your comments), in red (our comments).

Review – Nicu et al. 2022 „Multi-hazard susceptibility mapping of cryospheric hazards in a high-Arctic environment: Svalbard Archipelago"

I think the authors' work in Arctic context is very valuable as it contributes to filling a knowledge gap in an area affected by complex and understudied processes that have become an issue in light of climate change. Personally, I found the manuscript clear, well written, and well referenced.

We highly appreciate your positive general comment on our paper.

I do not see major issues but I tend to agree with the comment of the other reviewer calling for better stressing the limitations of the work. On my end, it seems to me that more stress was put into the modelling strategy and results rather than on the construction of the inventory which, in ESSD, should be the focal point. So, I suggest that the authors try to provide more details on the data collection, remote sensing imagery interpretation, and in-situ validation. I think that the GAM modelling, as one of the possible approach, can be shown as an example of potential use of the data but, I repeat, the focal point of the manuscript should be the presentation of the dataset itself.

As we already commented on the other reviewer's observations, we added some more details on the collection of data (L165-175). Our focus was also on the modelling strategy, but just to highlight what can be produced in terms of multi-hazard cryospheric modelling if comprehensible inventories (like our case) are available. More details were added in L426-432, regarding the data collection and interpretation.

---

## Author Response (AR2)

Dear Editor,

We did our best to correct all the typos in the final version of the manuscript. This was done by two of the authors. We hope this is a better version (without typos) than the previous one.

Kind regards,

I.C. Nicu on behalf of all authors